# Circular RNA as a Novel Biomarker and Therapeutic Target for HCC

**DOI:** 10.3390/cells11121948

**Published:** 2022-06-17

**Authors:** Hongwu Meng, Ruowen Niu, Cheng Huang, Jun Li

**Affiliations:** 1Inflammation and Immune Mediated Diseases Laboratory of Anhui Province, Anhui Institute of Innovative Drugs, School of Pharmacy, Anhui Medical University, Hefei 230032, China; 2046010052@stu.ahmu.edu.cn; 2Department of Pharmacy, The First Affiliated Hospital of Anhui Medical University, Hefei 230022, China; ayfynrw@163.com

**Keywords:** circular RNA (circRNA), alcoholic liver injury (ALI), hepatic fibrosis (HF), hepatocellular carcinoma (HCC), exosome

## Abstract

Circular RNA (circRNA) is a kind of endogenous non-coding RNA (ncRNA), which is produced by the reverse splicing of precursor mRNA (pre mRNA). It is widely expressed in a variety of biological cells. Due to the special formation mode, circRNA does not have a 5′ terminal cap and 3′ poly (A) tail structure. Compared with linear RNA, circRNA is more stable to exonuclease and ribonuclease. In addition, circRNA is structurally conserved, has a stable sequence and is tissue-specific. With the development of high-throughput sequencing and bioinformatics technology, more and more circRNAs have been found. CircRNA plays an important pathophysiological role in the occurrence and development of alcoholic liver injury (ALI), hepatic fibrosis (HF), hepatocellular carcinoma (HCC), and other liver diseases. Our group has been committed to the research of liver disease diagnosis and treatment targets. We review the function and mechanism of circRNA in ALI, HF and HCC, expecting to provide new ideas for the diagnosis, treatment, and prognosis of liver diseases.

## 1. Introduction

Circular RNA (CircRNA) was first discovered in plant viroids in 1976 [1]. In 1979, Hsu et al. discovered circRNA in eukaryotic cells using electron microscopy [2]. In 1993, circRNA was found in human cells [3]. Reportedly, circRNA is the product of wrong splicing. Next-generation sequencing has helped confirm the important role of circRNA in the occurrence and development of diseases.

Liver damage is one of the most important causes of morbidity and mortality worldwide. Liver diseases are common in China, including alcoholic steatohepatitis (ASH), non-alcoholic fatty liver disease (NAFLD), hepatic fibrosis (HF), hepatic cirrhosis (HC), and hepatocellular carcinoma (HCC) [4,5]. With the improvement in people’s living standards, fatty liver has become the second most prevalent liver disease after viral hepatitis. Fatty liver can develop into steatohepatitis without proper intervention. About 20–40% of steatohepatitis can develop into HF, HC, or even HCC [6]. This review systematically evaluates the role and mechanism of circRNAs in alcoholic liver injury (ALI), HF, and HCC, with a view to promoting the diagnosis and treatment of HCC involving circRNAs.

## 2. Biogenesis of circRNA

CircRNA is formed from pre mRNA by back-splicing. During the back-splicing process of circRNA, the downstream splicing donor is connected with the upstream splicing receptor to form a closed loop structure without a 5ʹ end cap and 3ʹ poly (A) tail [7]. CircRNA is not easily decomposed by RNA ribonuclease R (RNase R), and is more stable than linear RNA. CircRNA can be divided into three categories: exonic circRNA (ecircRNA), intron circRNA (ciRNA), and exon intron circRNA (EIciRNA). The formation of ecircRNA is mainly driven by “intron pairing-driven cyclization” and forms circRNA through reverse splicing [8]. About 80% of the discovered circRNAs come from exons (Figure 1).

## 3. Biological Functions of circRNA

CircRNA has unique structural and biological characteristics and is no longer regarded as a non-functional splicing “by-product”. It plays an important regulatory role in various diseases and has important pathological and physiological effects. Most circRNAs are expressed by known protein-coding genes and might consist of single or multiple exons. Although there is no 5ʹ cap and 3ʹ poly (A) tail, ecircRNA is usually present in the cytoplasm [9,10]. CiRNA and EIciRNA exist in the nucleus, and interact with U1 small nuclear ribonucleoprotein (snRNP, circEIF3J) [11] or positively regulate RNA polymerase II (pol II)-mediated transcription (ci-ankrd52) [12].

The functions of circRNA can be summarized as follows:

(1) CircRNA competitively binds to miRNA-binding sites, adsorbs miRNA similarly to a “sponge”, changes the abundance of miRNA inside or outside the cell, and regulates the expression of downstream target genes (Figure 2a). For example, as a sponge of miR-17-3p and miR-181-5p, cSMARCA5 can promote the expression of TIMP3 and inhibit the proliferation and migration of HCC cells [13].

(2) CircRNA can affect the function of RNA-binding protein. It has an advanced structure and can interact with RNA-binding protein (RBP) to affect mRNA expression (Figure 2b). For example, circAMOTL1 can combine with c-myc (cellular myelocytomatosis viral oncogene), STAT3 (signal transducer and activator of transcription 3), PDK1 (pyruvate dehydrogenase kinase isozyme 1), and AKT1 (protein kinase B 1) to promote their transfer to the nucleus, thus affecting the expression of the target genes [14]. Studies have shown that m6A modification regulates the circRNA-related immune response. CircFOREIGN can escape from self-detection by binding with YTHDF2 [15]

(3) CircRNA affects protein translation and, traditionally, it is a type of ncRNA. Therefore, circRNA might achieve its translation function through the internal ribosome entry site (IRES) and methylation of adenosine N6 (m6A) (Figure 2c). In eukaryotic cells, some circRNAs containing IRES can guide them to translate into peptides with biological functions. For example, circZNF609 can specifically regulate myoblast proliferation [16]. In prokaryotic cells, circRNA with an open reading frame (ORF) can be translated into proteins. For example, the green fluorescent protein (GFP) gene can be translated into GFP by circRNA [17]. Yang et al. found that N6 methyladenosine (m6A) is the most abundant base modifier of RNA, and m6A-driven circRNA translation is very common [18].

## 4. CircRNA and Liver Diseases

Studies performed on circRNA have confirmed its role in liver diseases.

### 4.1. CircRNA and Alcoholic Liver Injury

Alcoholic liver injury (ALI) is a liver disease caused by long-term excessive drinking. It often manifests as hepatic steatosis and high inflammatory cell infiltration [19]. Sustained development can lead to hepatocyte necrosis and even liver fibrosis. Currently, there is a lack of effective treatment except for alcohol abstinence and hepatoprotective drugs. In alcoholic hepatitis [15], our team found 20 differentially expressed circRNA, 11 downregulated and 9 upregulated, using high-throughput sequencing. In vitro cell experiments showed that mm9_circ_018725 may mediate the apoptosis and inflammation of AH [20]. Dou et al. found that there were 10 differentially expressed circRNAs, 4 downregulated and 6 upregulated, in the mouse model of alcoholic liver disease using RNA sequencing. Among them, mou_circ_1657 may affect the development of ALD through the interaction with miR-19-5b [21].

### 4.2. CircRNA and Hepatic Fibrosis

During hepatic fibrosis (HF), hepatic stellate cells (HSCs) are activated into myofibroblast-like cells that secrete a large amount of collagen, causing excessive deposition of extracellular matrix (ECM) and the formation of fibrous scar and inducing liver damage [22]. Interestingly, our previous research shows that HF is a reversible disease. When the pathogenic factors are eliminated, the course of HF can be reversed by itself [23]. However, further liver damage, such as cirrhosis and liver cancer, can be irreversible. Therefore, studying the pathogenesis of HF and finding potential therapeutic targets has garnered attention. In recent years, the important role of circRNAs in the clinical diagnosis and treatment of HF has been widely studied (Table 1). In activated HSCs and in mouse fibrotic liver tissues, upregulating circRNA circ-PWWP2A was suggested to promote HSC activation and proliferation via sponging miR-203 and miR-223, and subsequently promoting the expression of FSTL1 and TLR4 [24]. Our group has shown that overexpression of circFBXW4 can inhibit the activation and proliferation of HSCs, promote the apoptosis of HSCs, play the roles of anti-fibrotic factors, and alleviate liver inflammation. This effect may be related to the regulation of FBXW7 expression by the sponging of miR-18b-3p [25]. Another study showed that differentially expressed circRNA were closely related to liver oxidative stress, macrophage inflammation, and HSC activation in HF. In vitro cell experiment showed that mmu_circ_34116 inhibited the activation of mouse HSCs [26]. The inhibition of circ-0071410 increased the expression of miR-9-5p and inhibited HSCs’ activation [27]. CMTO1, a tumor suppressor gene, is downregulated in HCC. Some studies have found that cMTO1 mediates PTEN expression through miR-181b-5p, inhibits HSC activation, and produces an anti-fibrosis effect [28]. Circ608 can promote PINK1-mediated mitophagy of HSCs by acting as a sponge of miR222 in non-alcoholic steatohepatitis (NASH)-related liver fibrosis [29]. Hsa_circ_0070963 acts as an miR-223-3p sponge and inhibits the activation of HSC in HF by regulating LEMD3 [30]. In addition, hsa_circ_0004018 also inhibits the progress of HF by acting as an miRNA sponge. The hsa_circ_0004018/hsa-miR-660-3p/TEP1 axis can regulate HSCs’ activation and proliferation [31]. All these discoveries suggest that circRNA plays essential roles in the occurrence and progression of HF. However, only a very small proportion of circRNAs have been verified to regulate HF. Moreover, miRNA sponges are still the leading mechanism related to circRNAs in HF. Research into the exact correlation and potential mechanism between circRNAs and HF is still in its early stage.

### 4.3. CircRNA and Hepatocellular Carcinoma

Our knowledge of the pathogenesis of hepatocellular carcinoma (HCC) is incomplete. The effective treatment methods are surgery or transplantation, but the recurrence and metastasis rates are still high with a low 5-year survival rate [32]. Many circRNAs are involved in the proliferation, apoptosis, migration, and invasion of HCC cells. Therefore, circRNA may be an important biomarker for the clinical diagnosis and treatment of HCC.

According to the effect on the malignant phenotype of HCC cells, circRNAs can be divided into: carcinogenic circRNAs and tumor suppressor circRNAs. Compared with the adjacent tissues and normal liver cell lines, the expression of cancer-promoting circRNAs in HCC tissues and HCC cell lines is usually high. Inhibiting the expression of carcinogenic circRNAs can significantly reduce the proliferation, migration, and invasion of HCC cells, promote the apoptosis of HCC cells, and improve the prognosis. The phenotype of the tumor suppressor circRNAs is opposite to the above.

#### 4.3.1. Carcinogenic circRNAs

Circβ-catenin comes from the host β-catenin and is highly expressed in HCC. In vivo and in vitro experiments showed that silencing circβ-catenin significantly inhibited the malignant phenotype of HCC cells. This study shows that the so-called “non-coding RNA” actually has protein-coding ability. Circβ-catenin can be translated into a new subtype β-catenin-370aa, which promotes the progression of HCC by activating the Wnt/β-catenin pathway [33]. Sun et al. demonstrated that circ_0000105 is upregulated in HCC tissues and cells, and circ_0000105 can promote the proliferation and inhibit the apoptosis of HCC cells. Circ_ 0000105 can competitively bind miR-498 to promote the expression of PIK3R1. Circ_0000105 played an important role in tumorigenesis and development [34]. Wang et al. found that circ_10156 may act as a molecular sponge of miR-149-3p. Inhibiting circ_10156 upregulated miR-149-3p, downregulated AKT1 expression, and inhibited the proliferation of tumor cells [35]. Niu et al. showed that circ_0091579 is highly expressed in HCC tissues and cells, and promotes the proliferation and metastasis of HCC cells by downregulating the expression of miRNA-490-3p [36]. Qi et al. found that circ-PRKCI can reduce the expression of akt3, inhibit the apoptosis of HCC cells, and promote the invasion of HCC cells by sponging on miR-545. This study suggests that circ-PRKCI may affect the survival and prognosis of HCC patients by regulating E2E7 [37]. Fu et al. showed that circABCB10, as an miR-670-3p sponge, can upregulate the expression level of HMG20A. Therefore, the development of HCC can be blocked by inhibiting the expression of circABCB10 [38]. Huang et al. pointed out that the high expression of circRNA-100338 as an endogenous sponge of miR-141-3p promotes the invasion of hepatocellular carcinoma cells and is a potential biomarker for the diagnosis and treatment of HCC [39]. Hu et al. found that circASAP1 promotes the proliferation, migration, and invasion of HCC cells in vitro and promotes tumor growth and lung metastasis in vivo. Mechanism studies have found that circASAP1 can act on miR-326 and miR-532-5p to regulate the expression of MAPK1 and CSF-1 [40]. Wang et al. found that the high expression of circRHOT1 in HCC patients is associated with poor prognosis. The RNA pull-down assay showed that circRHOT1 introduced Tip60 into the NR2F6 promoter and started NR2F6 transcription. Knockout of NR2F6 inhibited the proliferation, migration, and invasion of HCC cells, while restoring NR2F6 expression in circRHOT1 knockout HCC cells rescued this effect. Therefore, pharmacological inhibition of NR2F6 expression may be a promising therapeutic target for HCC [41]. Li et al. found that circMAT2B can upregulate the expression level of PKM2 by sponging miR-338-3p, which encodes the key enzyme in the process of glycolysis and promotes the progress of glycolysis and HCC [42]. Lin et al. found that circular RNA Gprc5a can activate YAP1/TEAD1, a key downstream protein of the Hippo signaling pathway, through miR-1283, and promote the progress of HCC [43]. The above carcinogenic circRNAs and others promoting the malignant phenotype of HCC are shown in Table 2 [44,45,46,47,48,49,50,51,52,53,54,55].

The studies mentioned above show that carcinogenic circRNAs are usually upregulated in HCC, promoting the proliferation, migration, and invasion of tumor cells, mostly via sponging miRNAs.

#### 4.3.2. Tumor Suppressor circRNAs

CircTRIM33-12 is downregulated in HCC tissues and cell lines. The reduced expression of circTRIM33-12 was significantly correlated with malignant characteristics in HCC cells and promoted tumor proliferation, migration, invasion, and immune evasion. CircTRIM33-12 resulted in reduced 5-hydroxymethylcytosine (5hmC) levels in HCCs by sponging miR-191 to upregulate TET1 expression [56]. Hsa_circ_103809 negatively regulates the expression of miR-620. The decreased expression of hsa_circ_103809 promotes the proliferation, migration, and invasion of HCC cells [57]. Interestingly, Zhan et al. showed that hsa_circ_103809 was upregulated in HCC tissues and cell lines. Hsa_circ_103809 promotes the expression of FGFR1 by reducing the expression of miR-377-3P and promotes the cell cycle, proliferation, and migration of tumor cells [51]. Zhong et al. found that circC3P1 inhibits the growth and metastasis of HCC cells by upregulating the expression of PCK1 by sponging miR-4641 [58]. Song et al. showed that the expression of circADAMTS14 and RCAN1 was downregulated, and the expression of miR-572 was increased in HCC. CircADAMTS14 may inhibit the progression of HCC by regulating miR-572/RCAN1 as a competitive endogenous RNA [59]. CircADAMTS13 was also significantly downregulated in HCC tumor tissues. Mechanistically, circADAMTS13 directly interacts with miR-484 [60]. Zhu et al. showed that quaking (QKI) downregulated the expression of circZKSCAN1 in HCC and inhibited a variety of malignant manifestations by inhibiting the stemness of tumor cells. CircZKSCAN1 acts as a sponge of RNA-binding protein fragment X mental retardation protein (FMRP), competes with the cell cycle and apoptosis regulator 1 (CCAR1), and then inactivates the Wnt/β-catenin signaling pathway [61]. Sun et al. demonstrated that up-regulated circSETD2 strongly inhibited the proliferation, invasion and migration of liver cancer cells, whereas circSETD2 knockdown exerted the opposite effects [62]. Lin et al. showed that the expression of circCDK13 was decreased in HCC cell lines and patient tissues. Overexpression of circCDK13 inhibits the migration rate of tumor cells, alters cell cycle progression, and inhibits migration and invasion. CircCDK13 may play an anti-tumor role through JAK/STAT and PI3K/AKT signaling pathways [63]. Zheng et al. found that overexpression of hsa_circ_0079299 inhibited the growth of tumor cells and delayed the cell cycle progression in vivo and in vitro. The inhibitory effect of hsa_circ_0079299 on tumor cells may be mediated via the PI3K/AKT/mTOR signaling pathway [64]. Yu et al. found that circRNA-0072309 could inactivate PI3K/AKT and Wnt/β-catenin pathways by targeting miR-665 in Hep3B [65]. Another study found that circRNA cSMARCA5 inhibited the proliferation and migration of HCC cells in vivo and in vitro. CircRNA cSMARCA5 promotes the tumor suppressor TIMP3 expression by sponging miR-17-3p and miR-181b-5p [13]. Han et al. showed that the expression of circMTO1 was decreased in HCC tissues, reducing the survival rates of patients with downregulated circMTO1. CircMTO1 acts as a tumor suppressor by sponging miR-9 and promoting the expression of p21. This study suggests that circMTO1 is an important target for the treatment of HCC, and the decrease in circMTO1 in HCC may be an important factor for poor prognosis [66]. The above tumor suppressor circRNAs are shown in Table 3.

The reduced expression of tumor suppressor circRNAs is an important risk factor in HCC and has adverse effects on the proliferation, invasion, and metastasis of tumor cells. The phenotype and mechanism of these circRNAs provide new ideas for the clinical diagnosis of HCC. They can be a potential target for the treatment of liver cancer.

### 4.4. Exosome circRNA and Hepatocellular Carcinoma

Overexpressing circRNA Cdr1as in exosomes extracted from HCC cells promoted the proliferative and migratory abilities of normal cells, improving the expression of AFP by sponging miR-1270 [67]. Chen et al. found that circ-0051443 was transferred from normal cells to HCC cells through exosomes, and inhibited the malignant biological behavior of tumor cells by promoting apoptosis and blocking the cell cycle of HCC cells. Circ-0051443 regulates BAK1 expression through miR-331-3p [68] (Figure 3).

Lai et al. showed that circFBLIM1 was highly expressed in serum exosomes and HCC cells, and inhibited circFBLIM1 to limit glycolysis and HCC progression [69]. Huang et al. showed that the exosome-derived circRNA-100338 from MHCC97H promoted the invasion and metastasis of HCC cells [70]. Zhang et al. have shown that adipocyte-derived circ-deubiquitination (circ-DB) can inhibit miR-34a and activate ubiquitinated protein USP7, thus promoting tumor growth and inhibiting DNA damage [71]. Sun et al. demonstrated that high expression of hsa_circ_0004001, hsa_circ_0004123, and hsa_circ_0075792 can help diagnose HCC with greater sensitivity and specificity [72].

## 5. Summary and Prospect

CircRNAs are highly conserved and structurally stable, and play an important role in the occurrence and development of many diseases. We reviewed the current research on circRNAs in alcoholic liver injury (ALI), hepatic fibrosis (HF), and hepatocellular carcinoma (HCC). The purpose was to show the expression and mechanism of circRNAs in liver diseases, and provide a reference for the follow-up study of the role of circRNAs in liver diseases. However, there are several problems in the current research on circRNAs: (1) at present, the research on the mechanism of circRNAs is mostly focused on the sponging of miRNAs through the circRNA–miRNA–mRNA regulatory axis with few reports on other mechanisms. However, circular RNA is translatable, which can lead to the production of certain proteins. Encoding circular RNA may become a potential therapeutic target for liver cancer and become a new field of life science; (2) most research focuses on circRNAs and their downstream target genes with few studies on the upstream regulatory factors of circRNAs; (3) the naming of circRNAs is a little confusing. There are different names for the same circular RNA. In future research, we need to resolve the above problems to better clarify the biological function of circRNAs.

## Figures and Tables

**Figure 1 cells-11-01948-f001:**
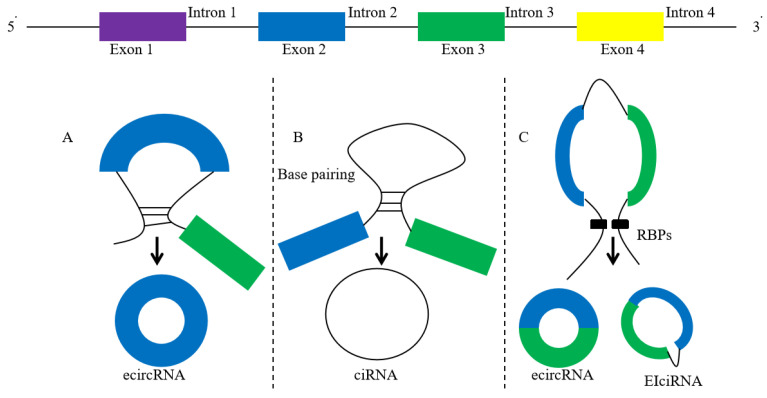
The characteristics of circRNAs and the models of their circularization (intron pairing-driven circularization). (**A**): exonic circRNA (ecircRNA), (**B**): intronic RNA (ciRNA), (**C**): circularization of ecircRNA and EIciRNA through RNA-binding protein (RBP).

**Figure 2 cells-11-01948-f002:**
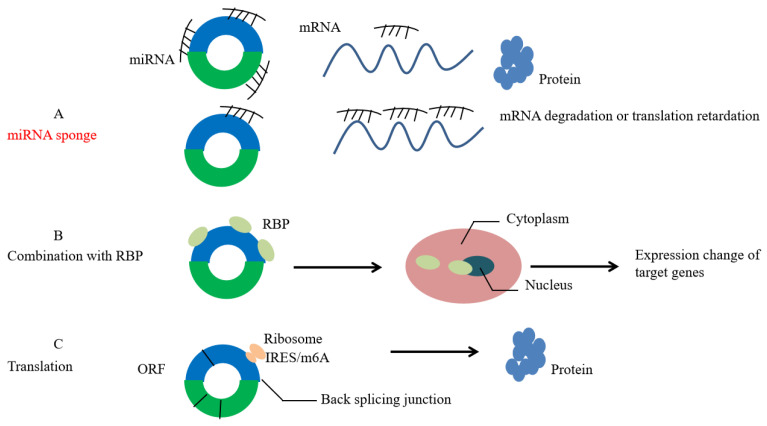
Function of circRNA. (**A**): circRNAs act as miRNA sponge. (**B**): circRNAs affect the function of RNA-binding protein (RBP). (**C**): circRNAs possess the function of translation by internal ribosome entry site (IRES) and methylation of adenosine N6 (m6A).

**Figure 3 cells-11-01948-f003:**
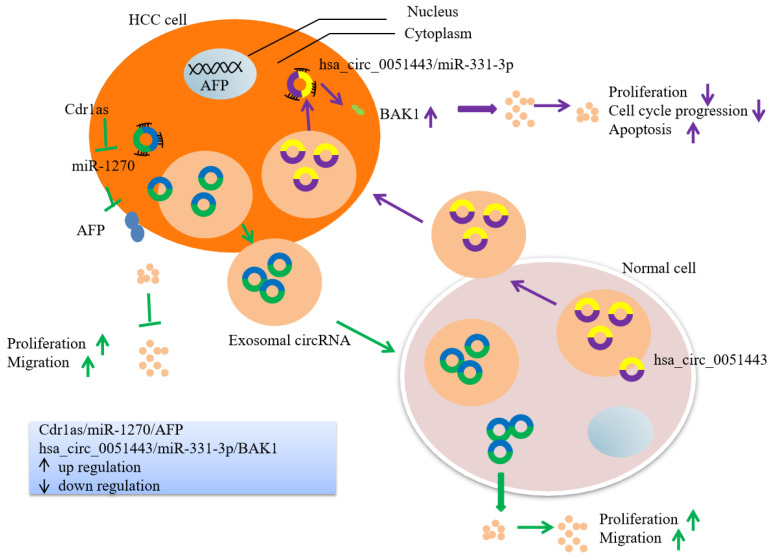
Exosomes transferred from HCC cells to surrounding normal cells or from surrounding normal cells to HCC cells mediate the biological progression of HCC. Exosomal circRNA Cdr1as stimulated malignant behaviors of surrounding normal cells. Exosomal circular RNA hsa_circ_0051443 suppresses the progression of HCC.

**Table 1 cells-11-01948-t001:** Expression and Function of dysfunctional circRNA in HF.

CircRNAs	Expression Change	Function	Possible Mechanism	References
hsa_circ_0074837	Up	HSC activation(+); proliferation(+).	miR-203/miR-223/FSTL1/TLR4	[24]
circFBXW4	Down	HSC activation(−); proliferation(−); apoptosis(+); Anti-inflammation.	miR-18b-3p/FBXW7	[25]
mmu_circ_34116	Down	HSC activation(−).	BMP7	[26]
hsa_circ_0071410	Up	HSC activation(+).	miR-9-5p/MRP1/ABCC1	[27]
hsa_circ_0007874/cMTO1	Down	HSC activation(−); proliferation(−).	miR-181b-5p/PTEN	[28]
circ608	Down	HSC mitophagy(+).	miR222/PINK1	[29]
hsa_circ_0070963	Down	HSC activation(−); proliferation(−); cell cycle(−).	miR-223-3p/LEMD3	[30]
hsa_circ_0004018	Down	HSC activation(−); proliferation(−).	miR-660-3p/TEP1	[31]

**Table 2 cells-11-01948-t002:** Expression and Function of carcinogenic circRNA in HCC.

CircRNAs	Expression Change	Function	Possible Mechanism	References
circβ-catenin	Up	Proliferation(+); migration(+).	β-catenin370aa/Wnt/β-catenin	[33]
circ_0000105	Up	Proliferation(+); apoptosis(−).	miR-498/PIK3R1	[34]
circ_10156	Up	Proliferation(+)	miR-149-3P/AKT1 pathway	[35]
circ_0091579	Up	Proliferation(+); migration(+).	miRNA-490-3p	[36]
circ-PRKCI	Up	Apoptosis(−); invasion(+).	miRNA-545/AKT3 and E2F7	[37]
circABCB10	Up	Proliferation(+); invasion(+).	miR-670-3p/HMG20A	[38]
hsa_circRNA_100338	Up	Biomarker for clinical diagnosis	miR-141-3p	[39]
circASAP1	Up	Proliferation(+); migration(+); invasion(+).	miR-326/miR-532-5p-MAPK1/CSF-1 signaling	[40]
circRHOT1	Up	Proliferation(+); migration(+); invasion(+); apoptosis(−).	TIP60-dependent NR2F6	[41]
circMAT2B	Up	Tumor-promoting; Glycolysis(+).	miR-338-3p/PKM2 axis	[42]
circGprc5a	Up	Proliferation(+); apoptosis(−).	miR-1283/ YAP1/TEAD1	[43]
hsa_circrna_100084	Up	Proliferation(+); migration(+); invasion(+).	miR-23a-5p/IGF2	[44]
circSOD2	Up	Cell growth(+); migration(+); cell cycle(+).	miR-502-5p/JAK2/STAT3	[45]
circHECTD1	Up	Proliferation(+); migration(+); invasion(+); apoptosis(−).	miR-485-5p/MUC1	[46]
circRNA_101237	Up	Diagnostic biomarker	Not shown	[47]
hsa_circ_0016788	Up	Proliferation(+); invasion(+); apoptosis(−).	miR-486/CDK4	[48]
circMET	Up	Invasion(+); metastasis(+); epithelial to mesenchymal transition (+); Immunosuppression(+).	miR-30-5p/snail/DPP4	[49]
circRNA-101368	Up	Migration(+).	miR-200a/ HMGB1/RAGE	[50]
hsa_circ_103809	Up	Proliferation(+); cell cycle(+); migration(+).	miR-377-3P/FGFR1	[51]
circHIPK3	Up	Proliferation(+); migration(+).	miR-124/AQP3	[52]
circ_104075	Up	Diagnostic biomarker and a therapeutic target.	HNF4a/circ_104075/ miR-582-3p/YAP	[53]
circZNF566	Up	Proliferation(+); migration(+); invasion(+).	miR-4738-3p/ TDO2	[54]
circTMEM45A	Up	tumor growth(+)	miR-665/IGF2	[55]

**Table 3 cells-11-01948-t003:** Expression and Function of tumor suppressor circRNA in HCC.

CircRNAs	Expression Change	Function	Possible Mechanism	References
circTRIM33–12	Down	Proliferation(−); migration(−); invasion(−); immune evasion abilities(−).	miR-191/TET1	[56]
hsa_circ_103809	Down	Proliferation(−); migration(−); invasion(−).	miR-620	[57]
circC3P1	Down	Proliferation(−); migration(−); invasion(−).	miR-4641/PCK1	[58]
circADAMTS14	Down	Proliferation(−); migration(−); invasion(−).	miR-572/RCAN1	[59]
circADAMTS13	Down	Proliferation(−)	miR-484	[60]
circZKSCAN1	Down	cancer stem cells(−)	Qki5/circZKSCAN1/FMRP/CCAR1/Wnt signaling axis	[61]
circSETD2	Down	Proliferation(−); invasion(−); migration(−).	E-cadherin, N-cadherin and Vimentin.	[62]
circCDK13	Down	Migration(−); cell cycle(−); invasion(−).	JAK/STAT and PI3K/AKT	[63]
hsa_circ_0079299	Down	Proliferation(−); cell cycle(−).	PI3K/AKT/mTOR	[64]
circRNA-0072309	Down	Tumor suppressor	miR-665/PI3K/AKT and Wnt/β-catenin pathways	[65]
hsa_circ_0001445	Down	Proliferation(−); migration(−).	miR-17-3p and miR-181b-5p/TIMP3	[13]
circMTO1	Down	Proliferation(−); invasion(−).	miR-9/p21	[66]

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
