# Peer review of "Circular RNA as a Novel Biomarker and Therapeutic Target for HCC"

_cells, 2022, doi:10.3390/cells11121948_

Round 1
Reviewer 1 Report
This is a fairly well written review of circRNAs in liver diseases that is reasonably easy to read. Minor typographical and grammatical changes needed in editing. Tables and figures suffer from lack of clarity due to poor font size and layout
Author Response
Thank you for your comments. We have polished the language of the manuscript through professional institution. For the lack of clarity in the tables and figures, we have adjusted the font and layout to provide a clearer version.
Reviewer 2 Report
This review deals with a largely new topic on which there are still sporadic papers published, sometimes without great links among them.
However, the Authors of the present manuscript were able to systematize the topic, managing all in all to create a fairly easy and fluent reading.
The work is contained in its length and is not excessive.
In section no. 4 (Summary and Prospect) the Authors critically analyze the result of their review. I think these considerations are appropriate.
The work contains 3 figures and 3 tables. These figures and tables are well done, didactic and contain appropriate content for the text.
As for my knowledge on the subject, the work seems to me well done and developed.
Author Response
Thank you very much for the comments. We will make persistent efforts in our future work.
Reviewer 3 Report
The topic of the review presented by Meng and co-authors is interesting and fits the focus of the journal. Overall, this review is well written and covers the most basic aspects of circRNA biology and their role in the pathogenesis of liver disease. The review will appeal to readers from different research fields and provides sufficient information for beginners to understand the field as well.
The only comment I have to make is that in both the abstract and summary the authors use the term "conserved." In the field of ncRNAs, conservation brings to mind miRNAs, which have identical sequences/functions in different organisms. However, while the number of miRNAs is at most a few thousand, the number of circRNAs could exceed a few hundred thousand, thus greatly increasing the complexity of the cellular transcriptome.
Therefore, authors should specify what they mean by conservation; is it conservation of sequence and function, or either. Please include relevant examples and appropriate references.
Author Response
Thank you for your comments. Some researches have pointed out that circRNA is very conservative in sequence. It is usually expressed in the form of circRNA in humans and mice, and sometimes it can be detected in Drosophila(1). However, other studies have found that the circRNA composed of introns does not have strong sequence conservation(2). Therefore, whether the conservation of circular RNA is caused by precise regulation due to its important function or by its own gene conservation remains to be further studied.
- Rybak - Wolf A,Stottmeister C,Glazar P,et al. CircularRNAs in the Mammalian Brain Are Highly Abundant,Con-served ,and Dynamically Expressed[J].Mol Cell,2015,58(5) :870 -885.
- Cocquerelle, C., Mascrez, B., Hetuin, D., and Bailleul, B. (1993) Mis-splicing yields circular RNA molecules. FASEB J7, 155-160
Reviewer 4 Report
The review is interesting. However, some improvements are needed.
Although the sub-topics are nicely organized, the content within each sub-topic needs to be more descriptive and organized better. The english needs some improvement too.
Information on the occurence and approaches to detect/identify circRNAs as well as the circumstances under which they are formed (if known) could be added in the section on Biogenesis in the review.
Certain sub-sections read like a list rather than a review of available literature. As mentioned above, the flow within these sub-sections needs to be improved.
Author Response
Thanks for your comments. We polished the language of the article. According to the requirements of the reviewers, we adjusted some statements in the article to meet the requirements for the publication of the review.